# Surface Displacement Measurements of Artworks: New Data Processing for Speckle Pattern Interferometry

Jessica Auber–Le Saux [1,2], Vincent Detalle [1,2,3,*], Xueshi Bai [1,4], Michalis Andrianakis [5], Nicolas Wilkie-Chancellier [2] and Vivi Tornari [5]

[1] C2RMF, Centre de Recherche et de Restauration des Musées de France, 14 Quai François Mitterrand, 75001 Paris, France
[2] SATIE, Systèmes et Applications des Technologies de l'Information et de l'Energie, CY Cergy-Paris Université, ENS Paris-Saclay, CNRS UMR 8029, 5 Mail Gay Lussac, 95031 Neuville sur Oise, France
[3] Institut de Recherche Chimie Paris, PSL Research University, Chimie ParisTech, CNRS UMR 8247, 75005 Paris, France
[4] Fondation des Sciences du Patrimoine/EUR-17-EURE-0021, 33 Boulevard du Port, MIR de Neuville, CEDEX, 95011 Cergy-Pontoise, France
[5] FORTH, Institute of Electronic Structure and Laser, Foundation for Research and Technology Hellas (IESL-FORTH), N. Plastira 100, 71110 Heraklion, Greece
* Correspondence: vincent.detalle@culture.gouv.fr

**Abstract:** Curators have developed preventive conservation strategies and usually try to control the temperature (T) and relative humidity (RH) variations in the museum rooms to stabilise the artworks. The control systems chosen by museums depend on the size and age of the building, the financial means and the strategies that can be adapted. However, there is a lack of methods that can monitor mechanical changes or chemical reactions of objects in real-time or regularly. It would therefore ideally be preferable to monitor each of them to alert them to preserve them. For this purpose, a non-destructive, non-contact, full-field technique, Digital Holographic Speckle Pattern Interferometry (DHSPI), has already been developed and allows direct tracking of changes on the surface of artworks. This technique is based on phase-shifting speckle interferometry and gives the deformation of the surface below the level of the micro-meter of the analysed object. In order to monitor the deformation continuously, a large number of images are acquired by DHSPI and have to be processed. The existing process consists of removing noise from the interferogram, unwrapping this image, and deriving and displaying a 2D or 3D deformation map. In order to improve the time and accuracy of processing the imaging data, a simpler and faster processing method is developed. Using Matlab®, a denoising methodology for the interference pattern generated during data acquisition is created, based on a stationary wavelet transform. The unwrapped image is calculated using the CPULSI (Calibrated Phase Unwrapping based on Least-Squares and Iterations) algorithm as it gives the fastest results among the tested methods. The unwrapped phase is then transformed into surface displacement. This process performs these steps for each interferogram automatically. It allows access to 2D or 3D deformation maps.

**Keywords:** digital speckle pattern interferometry; denoising; unwrapping; cultural heritage; preventive conservation; deformation maps

## 1. Introduction

Museums try to measure and then control the environment of their stored and exhibited artworks in order to ensure better preventive conservation. To control the environment of the artworks on display, they may regulate the number of visitors, open windows or use curtains. A common intervention for most museums is to install air conditioning in some of their specific rooms or display cases. However, as the diversity of materials and objects leads to different states of conservation, it can be difficult to ensure an overall

preventive environment and it is usually impossible to obtain direct information on the actual evolution of the material.

Until now, even though fluctuations in the museum environment are monitored and measured, the impact of these variations on works of art has not been studied and monitored, there is no tool available to conservators, curators and policymakers that can monitor works of art in the museum or outdoor environmental contexts. The conservation principles implemented are usually linked to temperature and relative humidity measurements, with a possible retroactive solution in the best case. In addition, the collections are regularly observed by competent personnel, but there are no measures to account for the regular evolution of the materials that make up the works of art. Thus, ideally, a technique that can directly monitor the works of art would be preferable as it would detect the first signs of changes or the appearance of induced defects in the materials.

A technology based on digital holographic speckle interferometry (DHSPI) has already been developed and has proven its ability to measure microscopic deformations such as layer detachment, voids, cracks, etc. [1]. This technique can be applied to any surface regardless of its size, shape, or material without any preparation. DHSPI detects invisible defects by their position, size, microstructure, propagation, and impact (invisible to the naked eye or below the surface). This is because the surface moves due to any changes in the overall body of the work. If there is a hidden defect in the depth of this object, it will eventually impact a displacement of the surface which will be detected using external excitation. However, users should be aware that this detection can be difficult depending on the size and depth as well as the dynamics of the defect, some defects are inactive such as knots in the wood, mostly when they are very well stabilised in the material. In this case, it seems that there is generally no impact on the conservation of the artwork. As the movement of the artwork's surface is a response to a natural or artificial external load, it can give qualitative and quantitative analyses of the artwork through environmental modifications. These modifications can be achieved by changing the surface temperature of the artwork, the relative humidity (RH) [2,3] of its environment or mechanical changes, or when using restoration processes such as laser cleaning with material removal [4].

The DHSPI technique [1,3,5–7] has recently demonstrated its ability to measure surface displacement during environmental fluctuations to measure the impact of environmental fluctuations on artworks. As the DHSPI only acquires variations of a fringe movement, the data recorded at different times of the measurement needs to be analysed and processed before information on the perpendicular displacement of the surface can be extracted. In this work, a new approach to data processing is presented in order to calculate the impact of these fluctuations on the artworks and focused on automating the data processing, which leads to speed optimization. First, the principles of the technique will be explained. Then, the proposed approach to data processing will be described. Finally, the protocol for optimising the data processing parameters will be presented by showing the results on different samples.

## 2. Model Specification of Data Processing

The DHSPI technique, based on the interference principle [8], is illustrated in Figure 1. Interferences are based on the wave properties of light. The propagation of light waves is periodic and this periodicity depends on the wavelength of the light. When two coherent light waves are superimposed with the same periodicity and intensity, they can interfere. There are two types of interference: in-phase and out-of-phase. In-phase interference means that the maxima of both waves and their minima occur at the same time, and the resulting intensity is twice the intensity of one wave. The phase difference is 0 [$2\pi$]. Out-of-phase interference occurs when the maximum of one wave is at the same time as the minimum of the second wave, so the resulting intensity is 0. The phase difference between these two waves is $\pi$ [$2\pi$].

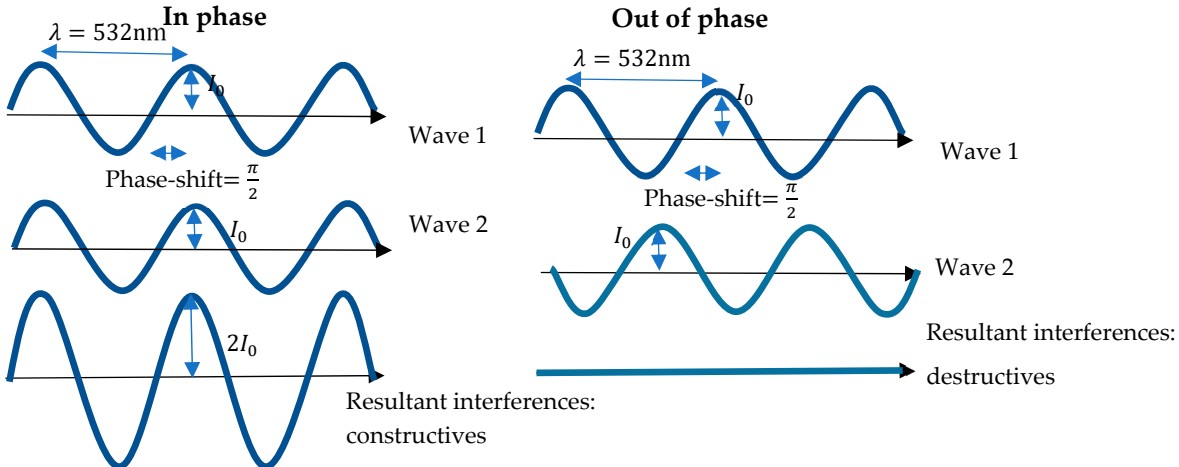

**Figure 1.** Interference principle.

*2.1. Pre-Processing*

The pre-processing step is the reconstruction of the interferogram [9–13]. For this purpose, one needs the images obtained in the initial state and the deformation state. On the one hand, the initial state can be the time of the beginning of the experiment, denoted $t_0$, and the deformation time is the set of other times of the DHSPI acquisitions, i.e., from time $t_1$ to time $t_n$ On the other hand, the initial state can be time $t_i$ and the deformation time the next one, the time $t_{i+1}$. The method is the same for both cases, but the second method is used in this experiment because if the technique were to be applied to a work of art in a museum, it would be logical to compare two images with the minimum time difference. During the acquisition of the DHSPI, five images of the surface of the object under study are recorded in order to optimise the signal-to-noise ratio. At time $t_i$, the pixel intensity of each image is formulated as:

$$I_n(x,y,t_i) = I_O(x,y,t_i) + I_R(x,y,t_i) + 2\sqrt{I_O(x,y,t_i)*I_R(x,y,t_i)}\cos\left[\varphi_R(x,y,t_i) - \varphi_B(x,y,t_i) - n*\frac{\pi}{2}\right] \text{ with } n=0,\dots,4 \quad (1)$$

where $I_O(x,y,t_i)$ is the object beam intensity at pixel $(x,y)$, $I_R(x,y,t_i)$ is the reference beam intensity, $\varphi_R(x,y,t_i)$ is the reference speckle pattern phase and $\varphi_B(x,y,t_i)$ is the phase of the undeformed sampling light beam.

$$I_n(x,y,t_{i+1}) = I_O(x,y,t_{i+1}) + I_R(x,y,t_{i+1}) + 2\sqrt{I_O(x,y,t_{i+1})*I_R(x,y,t_{i+1})}\cos[\varphi_R(x,y,t_{i+1}) - \varphi_B(x,y,t_{i+1}) - n*\tfrac{\pi}{2} + \Delta\varphi(x,y,t_{i+1})] \text{ with } n=0,\dots,4 \quad (2)$$

where $\Delta\varphi(x,y,t_{i+1})$ is phase difference that represents the surface movement between time $t_i$ and $t_{i+1}$.

In order to obtain the phase of time $t_i$, the images are recombined and deduced from Equation (1):

$$\tan[\varphi_R(x,y,t_i) - \varphi_B(x,y,t_i)] = \frac{2(I_2(x,y,t_i) - I_4(x,y,t_i))}{(2I_3(x,y,t_i) - I_1(x,y,t_i) - I_5(x,y,t_i))} \quad (3)$$

To obtain the phase of time $t_{i+1}$, the images are recombined and deduced from Equation (2):

$$\tan[\varphi_R(x,y,t_{i+1}) - \varphi_B(x,y,t_{i+1}) + \Delta\varphi(x,y,t_{i+1})] = \frac{2(I_2(x,y,t_{i+1}) - I_4(x,y,t_{i+1}))}{(2I_3(x,y,t_{i+1}) - I_1(x,y,t_{i+1}) - I_5(x,y,t_{i+1}))} \quad (4)$$

From Equations (3) and (4), the phase difference can be obtained:

$$\Delta\varphi = \tan^{-1}\left[\frac{2(I_2(x,y,t_{i+1}) - I_4(x,y,t_{i+1}))}{(2I_3(x,y,t_{i+1}) - I_1(x,y,t_{i+1}) - I_5(x,y,t_{i+1}))}\right] - \tan^{-1}\left[\frac{2(I_2(x,y,t_i) - I_4(x,y,t_i))}{(2I_3(x,y,t_i) - I_1(x,y,t_i) - I_5(x,y,t_i))}\right] \quad (5)$$

The phase difference is obtained by construction in the interval $[-\pi, \pi]$ thanks to the MatLab function, 'atan2' [14]. Therefore, the phase is wrapped and it needs to be unwrapped to obtain the surface deformation of the artwork.

Even if it is not always necessary, denoising the signal could enhance the quality of the data treatment.

### 2.2. Denoising Methodologies

Before unwrapping the interferogram, the image obtained with pre-processing technique had to be denoised. A simple and fast method coupled is sequentially applied: "fast denoising method based on the sine–cosine transform (SCT) and stationary wavelet transform (SWT) [15].

This simple and fast method uses a two-dimensional median filter. It is applied using the 'medfilt2' function in MatLab [16]. This function has three inputs: the image to be denoised, the window size, and the padding. The output pixel is the result of median filtering a given number of neighbourhood pixels by the window size. This window size is a square to obtain the same resolution in both dimensions. The padding is a way of extending the image boundaries, three options are offered: zero padding where the image is extended with zeros, symmetrical padding where the image boundaries are extended symmetrically, and indexed padding where the image extension is one.

The second method used to denoise the enveloped phase [15] shown in Figure 2 is a combination of the sine-cosine average (SCA) method [17,18] and the stationary wavelet transform method. In this method, firstly, the SCA (green in the figure) is the decomposition of the noisy interferogram into a sine and a cosine image. Then, these two types of images are recombined with the 'atan2' function after the SWT method. For each image (yellow parts in the figure), three steps are applied at this stage. An SWT transformation by the 'swt2' function is applied with a five-level decomposition and different analysis wavelets were considered. Before the application of this method, the image has to be expanded to meet the condition that the images have to be divided by $2^N$, where $N$ is the chosen level (in our case the condition is $2^5$). This is performed by the 'wextend' function with the periodic extension of the image with the 'per' mode. A Birgé-Massart threshold is defined with the function 'wbmpen' which takes as input an alpha, the tuning parameter for the penalty term, defined as [15] equal to 6.25 and a sigma, the standard deviation, obtained with 'wnoisest'. After the application of the threshold, the inverse SWT is applied by 'iswt2' and finally, the two images are recombined. To further denoise the wrapped phase, two different methods are used. The first denoising algorithm is a simple 2D median filter with a window size of $3 \times 3$ and symmetric filling. This method is followed by the SCA algorithm with SWT with a Birgé-Massart threshold, an alpha equal to 6.25, and a default sigma.

After this step, the image denoising is sufficient allowing us to continue the general data processing.

### 2.3. Unwrapping Process

After the denoising step, it is necessary to unwrap the phase. The method used in this work is the one recently described by Xia et al. [19]: Calibrated Phase Unwrapping based on Least-Squares Iteration (CPULSI). This method is applied using the MatLab algorithm created by the authors. The calibration method is based on least squares, iteration and calibration to phase derivatives. The iterations can be terminated by a minimum error reached or by a maximum number of iterations applied.

The MatLab function has several inputs: the image to be unwrapped, a mask that has the size of the phase to be unwrapped, and the pixels that are to be unwrapped have a value of 1 in the mask otherwise their values are 0 in the mask. Since unwrapping the whole image is required, our mask is a matrix containing only one. The third entry is the maximum number of iterations, followed by the minimum error to be achieved. The next inputs are the calibration and the coordinates of the known points of the phase. The

outputs of this function are the least squares unwrapped phase, the calibrated unwrapped phase, the actual number of iterations, and the computational time of the process. With this processing, a satisfactory unwrapped phase is obtained.

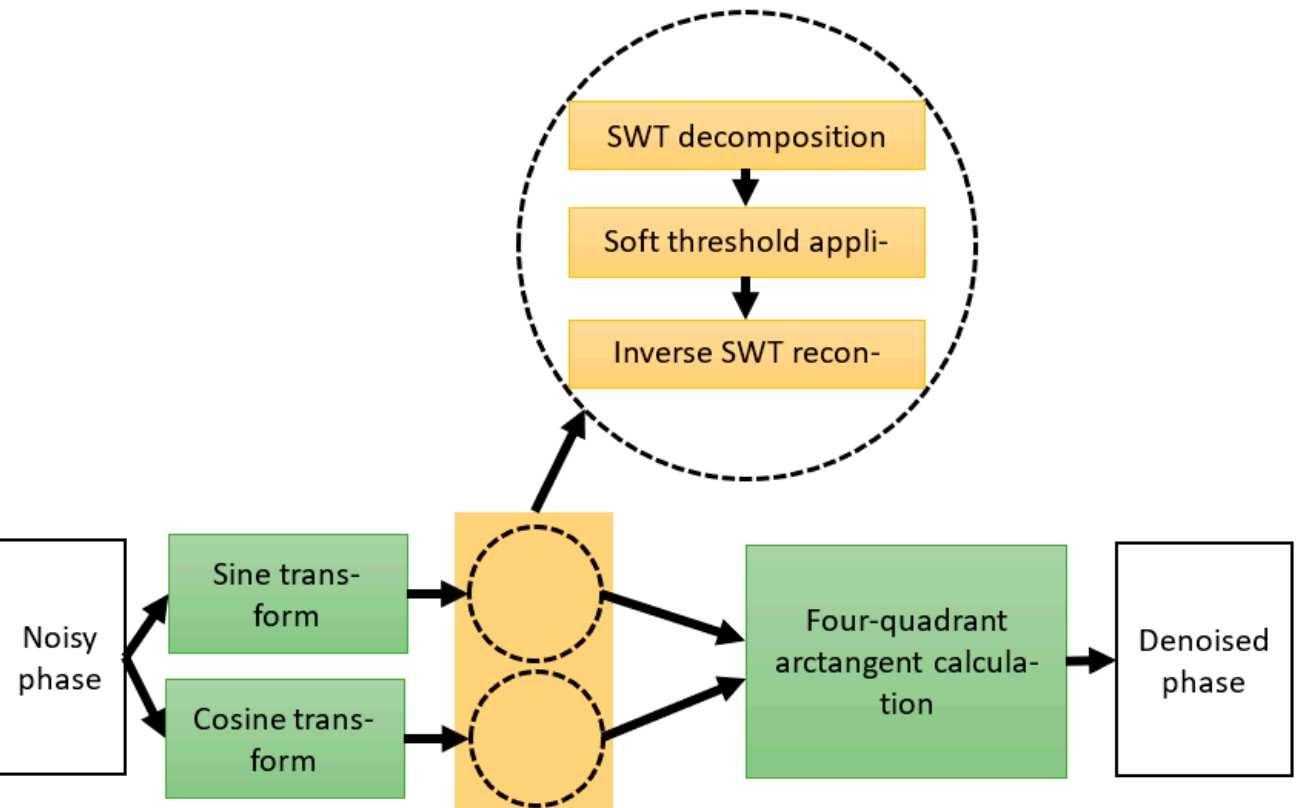

**Figure 2.** Scheme of the denoised method based on Sine-Cosine Average with Stationary Wavelet Transform principle. From the left to right, following the arrows, the different steps of the denoised methods are presented in different forms as different algorithms.

*2.4. Post-Processing*

To reach the deformation of the perpendicular variation of the surface, a post-processing algorithm should be applied. After the interferogram reconstruction, this image is denoised by median filtering with a window size of $3 \times 3$ and symmetric padding followed by the SCA-SWT denoising method. The experiments performed with DHSPI aim to find the deformation of the surface of objects that undergo height variation of the surface of the studied artwork. To measure this deformation, the DHSPI acquires images: the phase found in these images is of interest, hence the obtained denoised image needs to be unwrapped. In our processing procedure, the unwrapping is carried out by the CPULSI method. From the unwrapped phase, the surface deformation can finally be obtained: this step is called post-processing. The deformation is calculated for each pixel through:

$$d(x,y) = \frac{\lambda * \varphi(x,y)}{4\pi} \tag{6}$$

where $d(x,y)$ is the displacement of the surface at pixel $(x,y)$, $\lambda$ is the laser wavelength and $\varphi(x,y)$ is the calibrated unwrapped phase at $(x,y)$. The result of each pixel calculation is a deformation map that gives a quantitative displacement of the surface. The deformation maps can be drawn in 2D using the 'contourf' MatLab function or in 3D by the 'mesh' function.

To conclude, the overall processing used during the DHSPI acquisition consists of recording five images of a surface object at an initial time and five images of the same surface at a later time. During the time difference between the two acquisitions of the

image sets, the object undergoes a possible deformation. The whole data processing aims to quantify the deformation of the surface during this time difference. Several steps are necessary to obtain this deformation. The first one, called pre-processing, is the recombination of the interferogram, also called the wrapped phase because of the arctangent reconstruction. This is followed by processing, which includes denoising the interferogram and unwrapping it. Finally, post-processing is performed to calculate the deformation map to obtain the quantitative value needed to correlate the surface deformation of the object and the environmental fluctuations over time between two DHSPI acquisitions.

## 3. Materials and Experimental Parameters

### 3.1. Samples

In this article, the work has been focused on the wooden painting panels that are widely found in museums. A wooden panel has a complex structure (Figure 3): the support is made of wood, and then the preparation layer of gesso is mixed with glue composing the painting support. A pattern can be drawn directly on this layer, followed by the application of paint, which can sometimes be covered with a glaze mixture. A varnish is applied to the top of the wood panel [20] to protect the paint layer and increase the saturation of the colours [21].

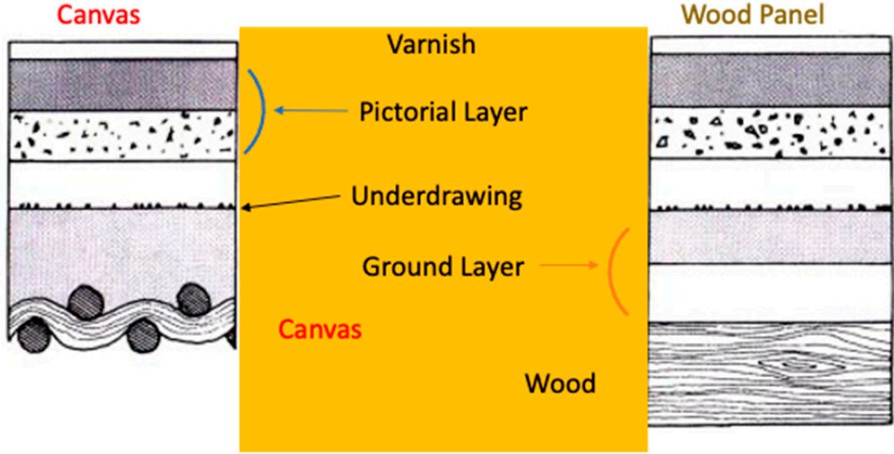

**Figure 3.** Typical wooden panel stratigraphy compares to easel painting.

Wooden panels are common artworks throughout history but each period has its peculiar artistic materials and implementations depending as well on the geographical localisation and cultural practices [22–24]. Hence, the focus of this paper is on the 15th and 16th centuries of European artists to narrow each type of layer especially pigments and wood support. From our known, artists in those centuries favoured hardwood species oak and poplar as for softwood species they preferred pine and spruce.

Therefore, spruce wood is chosen to perform the tested data proceeding. Two mock-ups made only of wood shown in Figure 4, were put through the experimental procedure. They are made with spruce, have the same dimensions of $100 \times 100$ mm and different thicknesses of 19 mm and 50 mm that are commonly used at this period. The impact of thickness should have different behaviour; thus, it is important to understand the impact of the volume of wooden support on the possible degradation.

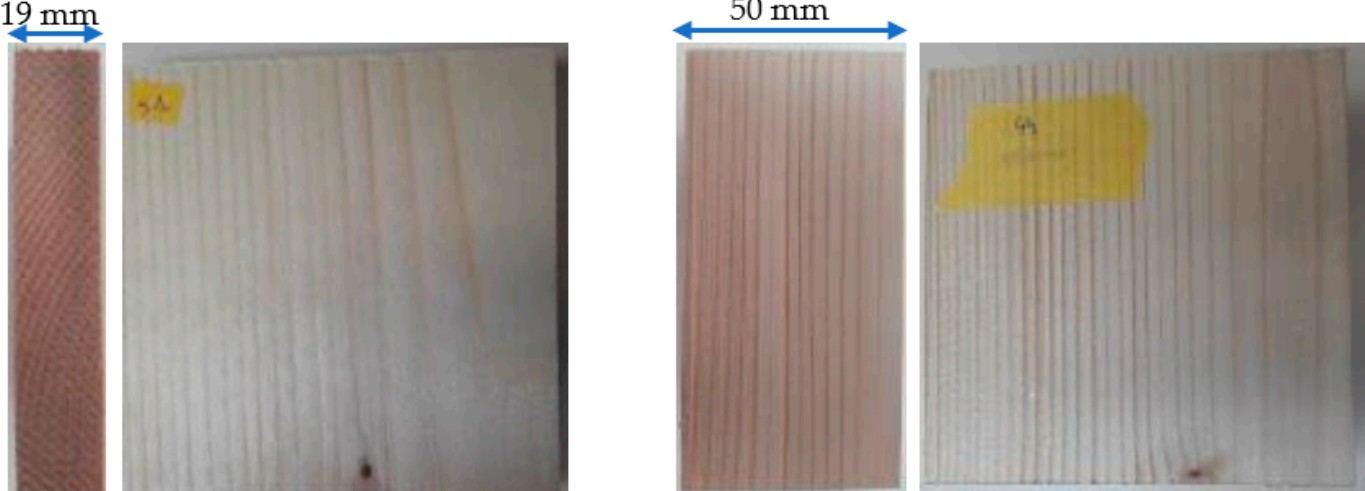

**Figure 4.** Sample 51 (**left**), spruce dimensions of $100 \times 100$ mm and a thickness of 19 mm. Sample 44 (**right**), spruce dimensions $100 \times 100$ mm and a thickness of 50 mm.

### 3.2. Experimental Parameters

The experiments were carried out in a homemade airtight climate chamber at FORTH and the aim was to evaluate the deformation of the surface under museum-like RH fluctuations. The DHSPI system has been developed by FORTH [1,6,7,25]. The optical path of the system is based on off-axis transmission holography in the double exposure phase-shifted holographic interferometry mode of application. The laser is a doubled frequency Nd:YAG, it has a wavelength of 532 nm and it is a continuous laser with a maximum power of 300 mW [7]. To improve the signal-to-noise ratio, a piezoelectric (PZT) mirror is used in the DHSPI with a $\frac{\pi}{2}$ phase shift. The detection system, CCD camera, allows getting a high resolution (2 Megapixels, 4.4 µm/pixel, $1600 \times 1200$ pixels) suitable for the intended application. Images were acquired automatically every five minutes for more than 2 days, this acquisition time is chosen by the slow variations induced. These data are available as ".bmp" images with pixel values in the interval of 0 and 255 thanks to the camera dynamics of 8 bits.

The airtight chamber is made of Plexiglas and equipped with environmental reading sensors and a scale for sample weight measurement in line with the recording DHSPI system. The sample, the environmental reading sensors and the scale are inside the custom-made climate chamber but the DHSPI system is outside the chamber. The chamber has a salt solution part where trays of saturated salt solutions are placed. The vapour of the saturated salt solutions enters the main chamber through holes distributed all over to allow for a simultaneous homogenisation of RH in the main chamber. To achieve low relative humidity variations in slow rates of change, lithium chloride saturated solutions to decrease the RH of the chamber and water vapour to increase the chamber's RH were used. The experiments were planned as follows: (a) shrinkage procedure, decrease from 75% to 25% RH, (b) swelling procedure, an increase of RH from 25% to 75% RH.

The images of the camera are cropped to obtain the entire surface of the sample.

## 4. Results

The general data processing is applied as presented previously to measure the impact of the RH fluctuations on both samples of spruce of thickness 19 mm and 50 mm. For one sample, the amount of data is more than 500 images to be processed.

The data acquired by DHSPI are presented in Figure 5 from image (a) to image (e) for time $t_{17}$ and images from (f) to (j) after five minutes at the time $t_{18}$. With Equation (5) is obtained image (k), the interferogram between time $t_{17}$ and $t_{18}$.

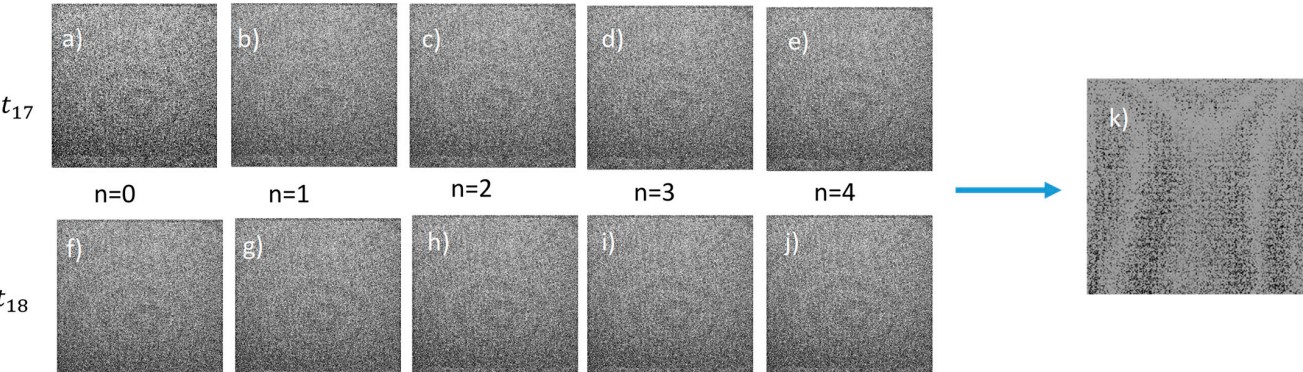

**Figure 5.** Pre-processing result: interferogram between time t = 85 min and time t = 90 min of sample n°51. Data acquired by DHSPI from image (**a–e**) for time $t_{17}$ and images from (**f–j**) after five minutes at the time $t_{18}$. Image (**k**) is the interferogram between time $t_{17}$ and $t_{18}$ obtained from Equation (5).

Between $t_{17}$ and $t_{18}$, there is a variation from 72% to 71%. A first pattern could be identified for these five minutes. Then, the impact of higher RH variations is followed during all the experimenting time.

Then, the efficiency of the Matlab denoising median filter, 'medfil2', is evaluated, as its different parameters, the padding and the window size. The three different padding available in the Matlab library were tested on the interferogram of sample 51 between the 85th min and 90th min, with the default window size of 3 by 3, the results are visible in Figure 3. Image (a) is the zeros padding, (b) is the symmetric padding and (c) is the indexed padding. Some of the main differences between the three paddings are represented by a circle with different colours, respectively.

In Figure 6 images (a) and image (c) rounded in green some remaining dots represent noise that are results and the image. It can be deduced that less noise is seen in image (b), therefore, the symmetric padding is the best padding for our median filter. It appears with this first data treatment evaluation that the choice of symmetric padding seems to be the most efficient for these experimentations and these recording conditions.

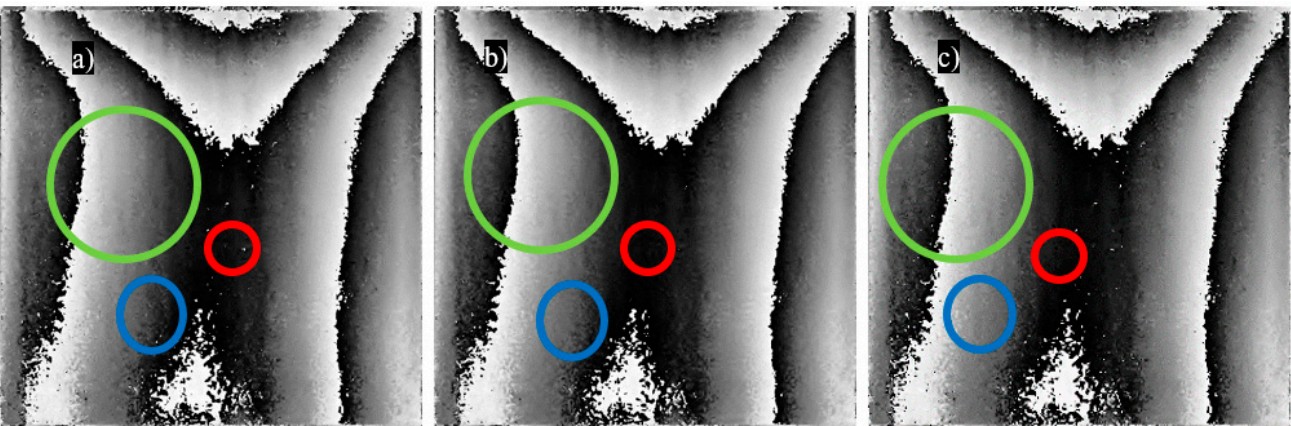

**Figure 6.** 'medfilt2' different with window default window size padding results on interferogram of sample 51 between 85th min and 90th min: (**a**) zeros padding, (**b**) symmetric padding and (**c**) indexed padding. The circles represent the main difference in the three images, the image (**b**) with the symmetric padding gives less noise.

With this padding, one needs now to determine the best square window size allowing us to optimise the details determination and the most efficient contrast visualisation. The denoised image by the median filter, 'medfilt2' by Matlab, is determined by the window size and has been tested from $2 \times 2$ to $7 \times 7$. The results of these tests are presented in

Figure 7 with symmetric padding on the interferogram of sample 51 between the beginning of the experiment and five minutes later for images (a) to (f) and on the interferogram of sample 44 between the images acquired five minutes after the beginning of the experiment and the set of images taken after 10 min of the experiment's beginning for images (g) to (l).

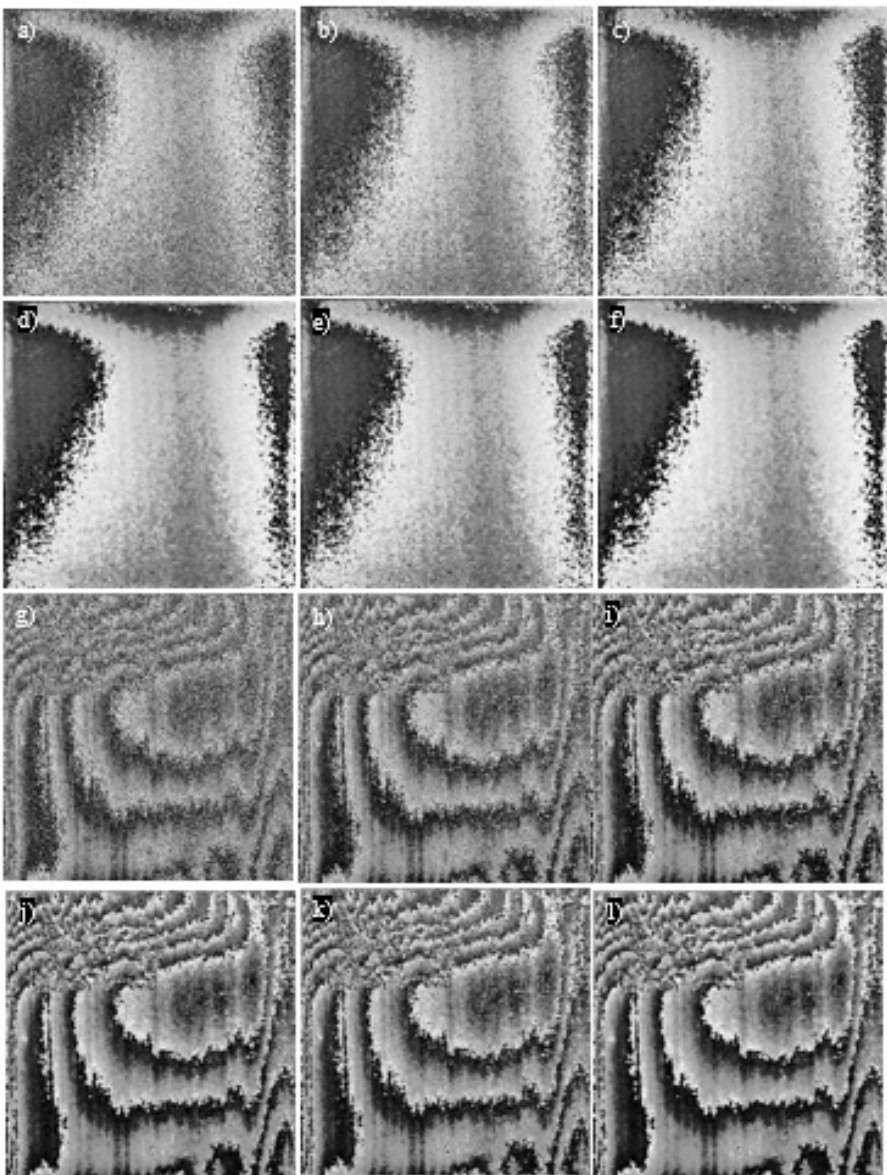

**Figure 7.** Denoising with Matlab function medfilt2 with symmetric padding. Interferogram of sample 51 between the beginning of the experiment and five minutes later denoised by a median filter with a window size of (**a**) 2 × 2, (**b**) 3 × 3, (**c**) 4 × 4, (**d**) 5 × 5, (**e**) 6 × 6 and (**f**) 7 × 7. The interferogram of sample 44 between the images acquired five minutes after the beginning of the experiment and the set of images taken after 10 min of the experiment with a window size of (**g**) 2 × 2, (**h**) 3 × 3, (**i**) 4 × 4, (**j**) 5 × 5, (**k**) 6 × 6 and (**l**) 7 × 7.

The more the window has a larger size (7 × 7), the more the contrast is important. However, at the same time, an increase can be observed in the fading of details. Then, the best relevant compromise needs to be found for the aims and application of this work.

From image (b), the grown rings can be observed, which is not possible in image (a). The window size of 2 × 2 does not denoise enough. From image (j), the details start to fade so the window size is too large from 5 × 5. The larger the window is, the more blurred areas appear. The best windows for our application are 3 × 3 and 4 × 4, shown by images

(b), (c) and (h), (i) in Figure 7. The time of computing is slightly increasing with the increase of the size of the window. But this computing time stays under 0.15 s, a mean time of 20 images, which is quite negligible. For the chosen window size of $3 \times 3$, the computing time is 0.04 s.

Then, the second possible denoising algorithm SCA-SWT is tested. The method has been tested first on the raw interferogram Figure 8 image (a) and image (c) and on the interferogram denoised by the MatLab median filter with symmetric padding and the $3 \times 3$ window Figure 8 images (b) and (d). The second denoising algorithm was not preceded by any previous denoising in the article of Ning et al. [15]. Images (a) and (b) are tests on the interferogram of sample 51 between the beginning of the experiment and five minutes later whereas images (c) and (d) are tests on the interferogram of sample 44 between the beginning of the experiment and five minutes.

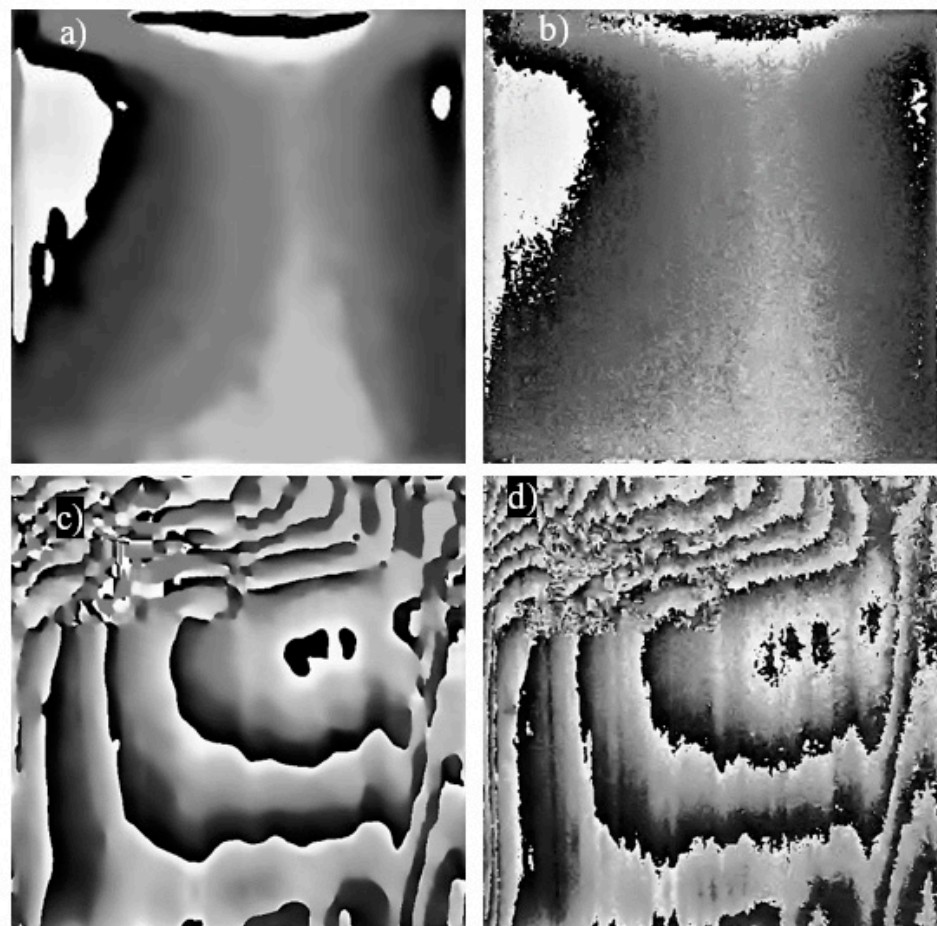

**Figure 8.** SCA-SWT denoising algorithm on interferogram of sample 51 between the beginning of the experiment and five minutes later (**a**) without any previous filter and (**b**) previously denoised by a median filter with symmetric padding and a window size of $3 \times 3$, interferogram of sample 44 between the beginning of the experiment and five minutes later (**c**) without any previous filter and (**d**) previously denoised by a median filter with symmetric padding and a window size of $3 \times 3$.

It appears clearly that without previous denoising operated by a median filter with symmetric padding and a window size of $3 \times 3$ one has lost too many details that will induce a lack of information.

For this data treatment using the SCA-SWT, the work is performed with a wavelet db4 indicated in the article by Ning et al. [15]. However, different analysis wavelets can be investigated and it is needed to optimise the different possible parameters.

To that end in Figure 9, the different kinds of wavelets were tested on an interferogram of sample 51 between 15th min and 20th min after the beginning of the experiment for images (a) to (d) and (i) to (l), the same processing was applied on interferogram of sample 44 between 5th min and 10th min after the beginning of the experiment for image (e) to (h) and (m) to (p). The results applied to the two mock-ups are shown with the shape of the wavelet. Before using the denoising process, the interferograms are denoised with the Matlab function 'medfilt2' with parameters symmetric padding and a window size of $3 \times 3$. The computing time is increasing with the increase of the peak number of the analysing wavelet. Two kinds of wavelets data treatment have been tested and compared Daubechies (db) and Symlets (sym) analysing wavelets to identify if our methodology is most relevant for the experiment which has been performed, the other parameters of the method are a five-level decomposition, an alpha equal to 6.25 and a Birgé-Massart threshold.

The SCA-SWT method takes less time with a db2 analysing wavelet than with a db4, but the computing time has a maximum of less than a second and a half for each image. For 564 images for processing of mock-up n°51 data, the processing takes from 8 min and 16 s for sym2 to 11 min and 43 s for sym8, and for 582 images for processing of sample 44 data, it takes 8 min and 42 s for sym2 to 11 min and 25 s for db8. The different images in Figure 9 do not seem to have that much of a difference. There is some information lost when the analysing wavelet is db4 or higher and sym4 or higher, this observation can be explained by the wavelet form. Indeed, the db2 and sym2 wavelets are sharper, which corresponds to the sharp edges found in the interferogram. Therefore, the db2 and sym2 analysing wavelet are possible for the expected application.

The border extension of the image during this method can bring some problems that are rounded in Figure 9 image (a). This image is the denoised interferogram of the beginning of the experiment and five minutes later sample 51, the image has a discontinuity on its border that can induce false displacement on the sides of the images. No discontinuities are visible for the other sample images. Therefore, the algorithm drawbacks depend on the processed interferogram and cannot be predicted.

After the last step, the unwrapping method is tested based on the CPULSI algorithm including the evaluation of the limit suitable iterations and minimising the error. The value ite$_{Max}$300e0.001 has a suitable limit at 300 iterations and an error to reach 0.001. The phase unwrapped by FORTH software is also plotted to compare our results. The CPULSI algorithm did not unwrap a denoised image in the article of Xia et al. [19], this method is tested. The Phase Unwrapping based on Least-Squares (PULSI) results available by the Matlab function are also presented, this algorithm presents the advantage of not using calibration and is suitable for denoised images. The CPULSI algorithm needs a phase-known point that is chosen as the point in the raw or denoised interferogram that has the phase nearer to zero taking the minimum in absolute value in the entire image. The denoised images are obtained with a median filter with symmetric padding and a window size of $3 \times 3$, followed by the SCA-SWT algorithm with a db2 analysing wavelet, in Figure 10 interferograms between the beginning of the experiment and five minutes later of experiment on sample n°51 are used whereas in Figure 11 interferogram between five minutes after the beginning of the experiment and five minutes later on sample n°44 are tested.

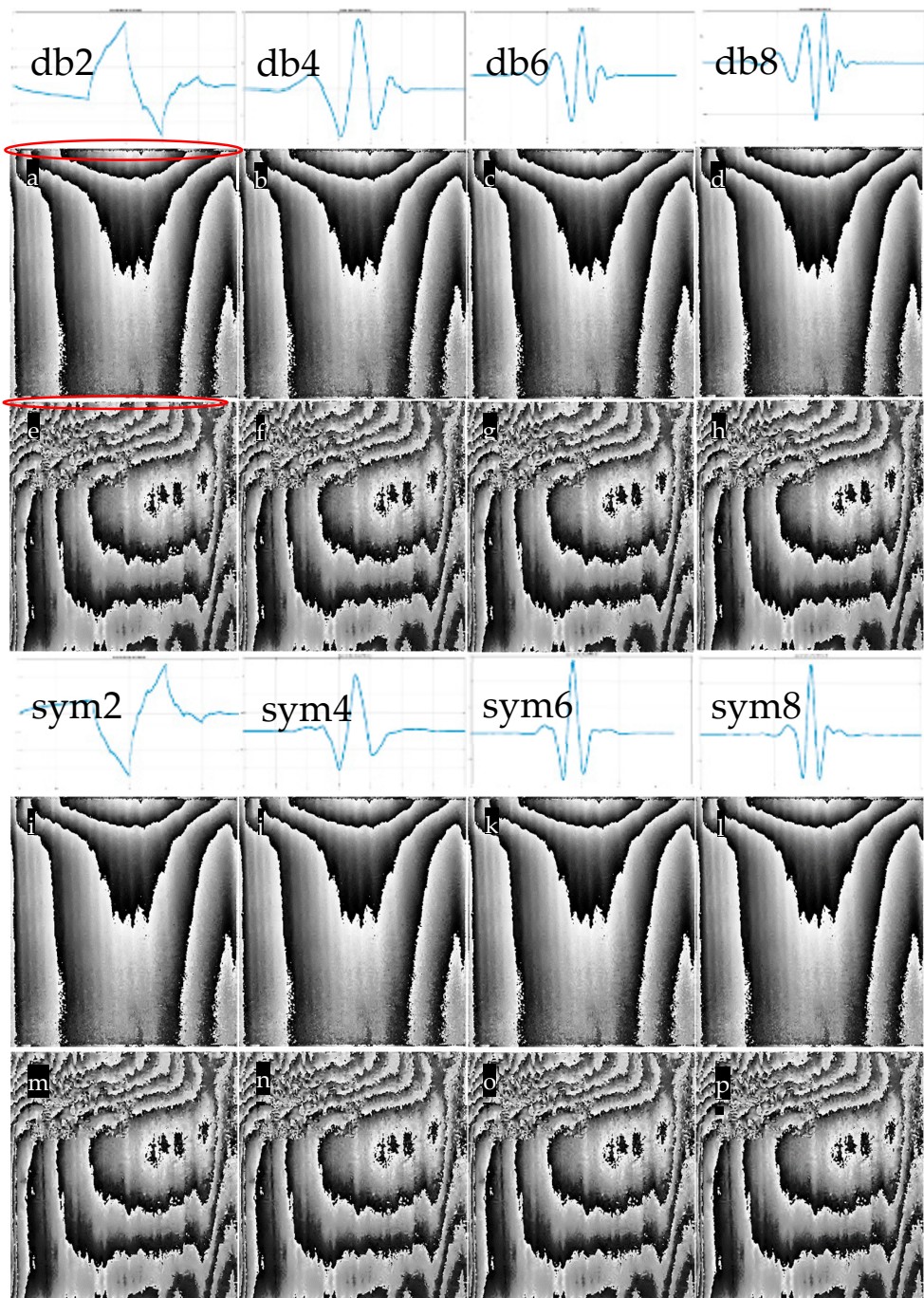

**Figure 9.** Denoising with Matlab function 'medfilt2' with symmetric padding and window size $3 \times 3$ followed by the algorithm SCA-SWT with a five-level decomposition, a default sigma found through 'wnoisest', an alpha equal to 6.25 and a Birgé-Massart threshold. Interferogram of sample 51 between 15th min and 20th min after the beginning of the experiment with (**a**) a db2 analysing wavelet, (**b**) a db4 analysing wavelet, (**c**) a db6 analysing wavelet and (**d**) a db8 analysing wavelet. Interferogram of sample 44 between 5th min and 10th min after the beginning of the experiment with (**e**) a db2 analysing wavelet, (**f**) a db4 analysing wavelet, (**g**) a db6 analysing wavelet and (**h**) a db8 analysing wavelet. Interferogram of sample 51 between 15th min and 20th min after the beginning of the experiment with (**i**) a sym2 analysing wavelet, (**j**) a sym4 analysing wavelet, (**k**) a sym6 analysing wavelet and (**l**) a sym8 analysing wavelet. Interferogram of sample 44 between 5 min and 10 min after the beginning of the experiment, (**m**) a sym2 analysing wavelet, (**n**) a sym4 analysing wavelet, (**o**) a sym6 analysing wavelet, (**p**) a sym8 analysing wavelet.

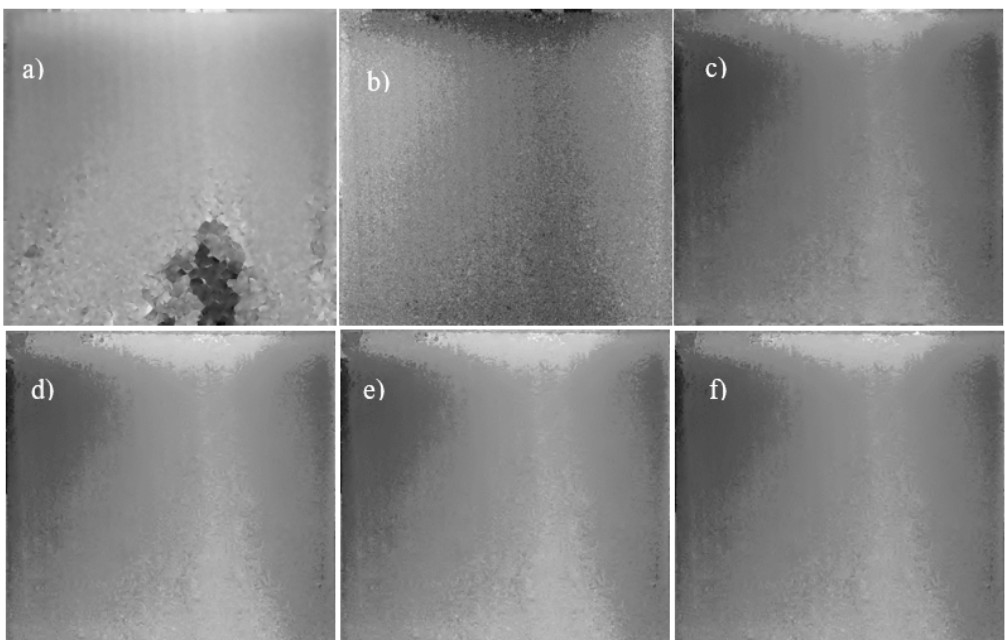

**Figure 10.** The unwrapped phase of interferogram between the beginning of the experiment and five minutes later of experiment on sample n°51 with (**a**) FORTH software, (**b**) CPULSI algorithm without denoising ite$_{Max}$300e0.001, (**c**) PULSI algorithm applied on denoised interferogram with ite$_{Max}$300e0.001, (**d**) CPULSI algorithm applied on denoised interferogram with ite$_{Max}$300e0.001, (**e**) CPULSI algorithm applied on denoised interferogram with ite$_{Max}$600e0.001 and (**f**) CPULSI algorithm applied on denoised interferogram with ite$_{Max}$300e0.0001.

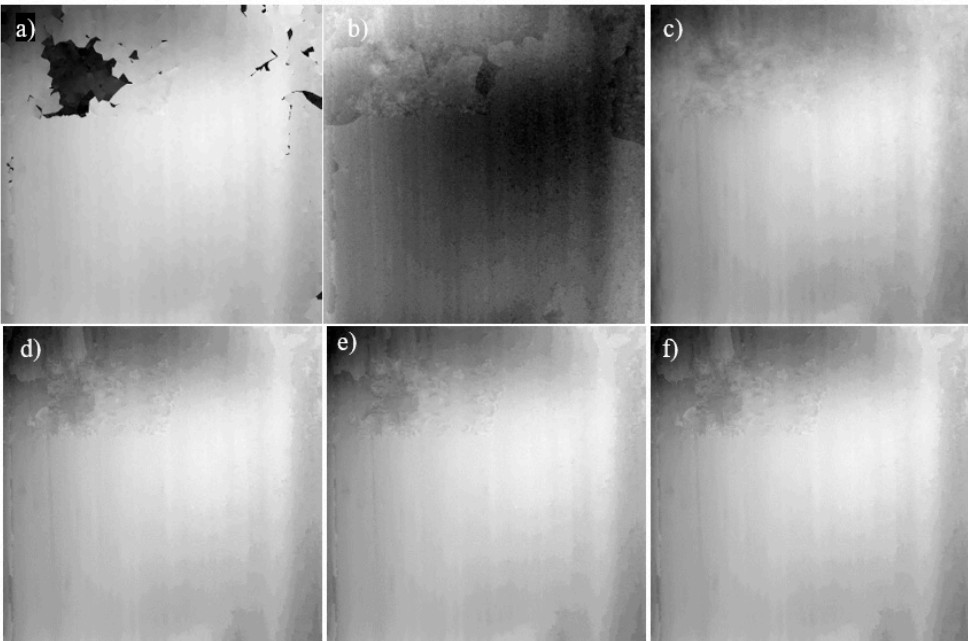

**Figure 11.** Unwrapped phase of interferogram between five minutes after the beginning of the experiment and five minutes later on sample n°44 with (**a**) FORTH software, (**b**) CPULSI algorithm without denoising ite$_{Max}$300e0.001, (**c**) PULSI algorithm applied on denoised interferogram with ite$_{Max}$300e0.001, (**d**) CPULSI algorithm applied on denoised interferogram with ite$_{Max}$300e0.001, (**e**) CPULSI algorithm applied on denoised interferogram ite$_{Max}$600e0.001 and (**f**) CPULSI algorithm applied on denoised interferogram with ite$_{Max}$300e0.0001.

These images need to be unwrapped to obtain a displacement of the surface. Figure 10 image (a) and Figure 11 image (a) are obtained with FORTH software [7]. The result of our interferogram unwrapped without denoising is shown in Figure 10 image (b) and Figure 11 image (b) obtained with the CPULSI algorithm with a suitable limit of iterations at 300 and an error to reach 0.001 (ite$_{Max}$300e0.001). The computing time is on average for the processing of 40 images, 41 s by image. The maximum and minimum phases are reversed concerning the images obtained by FORTH. The PULSI algorithm results on our images are presented in Figure 10 image (c) and Figure 11 image (c) with ite$_{Max}$300e0.001, the computing time is on average 0.51 s. The phases maximum and minimum are in correlation with FORTH results. The images (d) of Figures 10 and 11 are obtained with the CPULSI algorithm with ite$_{Max}$300e0.001. The computing time is on average 28.41 s. The computing time is shorter when denoised algorithms are used and the images are less noisy. Figure 10 image (d) and Figure 11 image (d) is obtained with the CPULSI algorithm with ite$_{Max}$600e0.001. The computing time is 45.71s, it is increased and there is not a visible improvement from the number of iterations of 300. Indeed, for some unwrapped phases, the error was reached with less than 300 iterations as in Figure 10 images (d) and (e) that are the same. Nevertheless, Figure 11 images (e) and (f) are different because the 300 iterations were reached and not the error. Figures 10f and 11f have inputs of ite$_{Max}$300e0.0001. Figure 11 images (e) and (f) are the same because the error of 0.001 was not reached with 300 iterations, a smaller error cannot be reached without any more iterations, and the computing time is 30.65 s. The CPULSI inputs chosen for our experiment is 300 maximum iterations for an error to reach 0.001.

The inputs of the function depend on the denoised interferogram. Indeed, Figure 10 is the unwrapped phase of Figure 9 image (a) that has fewer fringes than Figure 9 image (e), they are the chosen denoised images of mock-up n°51 and mock-up n°44 respectively to be unwrapped with CPULSI algorithm. To unwrap these two different images, the error to reach is set to 0.001 and the number of iterations is equal to 300. Figure 9 image (a) unwrapping ends because the error 0.001 is reached, contrary to the unwrapping of Figure 9 image (e) which unwrapping ends because the 300 iterations are reached. To improve the unwrap image, for image (a) the error to reach has to be improved, as shown in Figure 10 image (f) but for Figure 9 image (e) the number of possible iterations has to be improved as shown in Figure 11 image (e). The conclusion can be made that the inputs of the function depend on the obtained interferogram and that if more fringes are seen, more iterations will be needed.

Depending on what one is looking for, it is needed to choose one or the other unwrapping algorithm. The first one PULSI can give us in 0.5 s the absolute variation or displacement of one image useful for giving an alert. The second one CPULSI with a longer calculation time, around 30 s for one image, will give the real value and orientation of the displacement.

The final processing is the calculation of the displacement. Figure 12 shows the deformation map of the two mock-ups obtained during the experiment. Images (a) from (c) are deformation maps between the beginning of the experiment and five minutes later inthe experiment on sample n°51 whereas images (d) to (f) are Deformation maps between five minutes after the beginning of the experiment and five minutes later on sample n°44. The deformation map is obtained with a denoised image by a median filter and the SCA-SWT algorithm and unwrapped with a CPULSI algorithm with 300 maximum iterations and an error to reach 0.001 with a phase-know point is chosen as the point in the denoised image that has the phase nearer to zero.

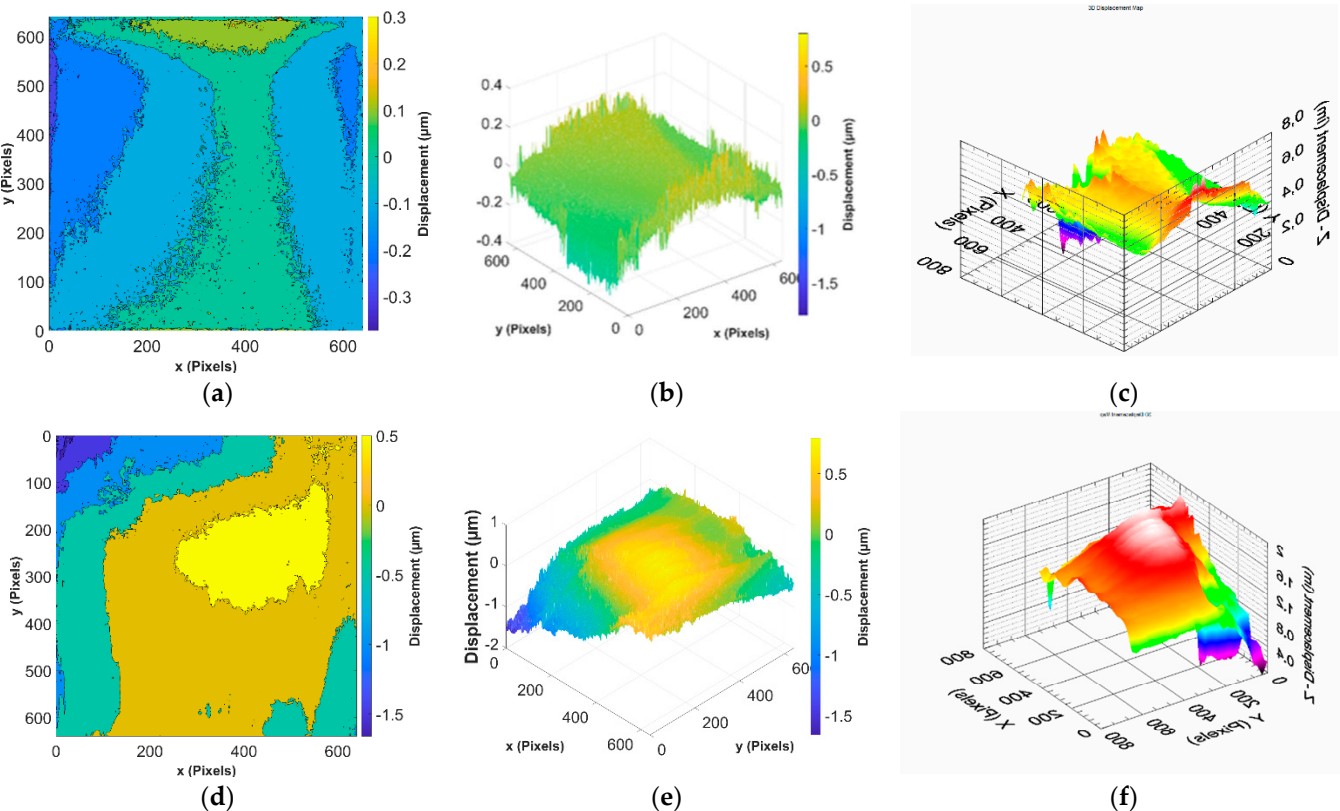

**Figure 12.** Deformation maps between the beginning of the experiment and five minutes later of experiment on mock-up n°51 (**a**) 2D of the unwrapped phase of interferogram with CPULSI algorithm applied on denoised interferogram with ite$_{Max}$300e0.001, the phase–known point at the point nearer to zero in the denoised interferogram (**b**) 3D of the same unwrapped phase (**c**) FORTH processing. Deformation maps between five minutes after the beginning of the experiment and five minutes later on mock-up n°44 (**d**) 2D of the unwrapped phase of with CPULSI algorithm applied on denoised interferogram with ite$_{Max}$300e0.001, the phase–known point at the point nearer to zero in the denoised interferogram (**e**) 3D of the same unwrapped phase (**f**) FORTH software.

After the reconstitution of the interferogram, the obtained wrapped phase is denoised using two different algorithms. First, a native 2D median filter of MatLab is used with a window size of $3 \times 3$ and symmetric padding. Then SCA-SWT method is applied with a db2 analysing wavelet, a Birgé-Massart threshold, an alpha equal at 6.25, and a default sigma. The denoised entire image is unwrapped by a CPULSI MatLab function based on a CPULSI algorithm with a phase-known point chosen as the point in the denoised image with the value nearer to zero, a suitable limit of iterations at 300 and an error to reach 0.001. The user can choose to visualise the deformation in 2D or 3D. All these steps are automatised and all the acquired images can be processed in one row.

## 5. Discussion

Experiments of RH variations have been carried out on softwoods that are representative materials used for panel paintings between the 15th and 16th centuries, inside an environmental chamber simulating real variations found inside specific museums. The homemade airtight climate chamber helps to stimulate the correlation between the RH fluctuations with the surface deformation of the object of interest. The RH fluctuations followed in the climate chamber are based on RH variations found in a museum's rooms. Two mock-ups made of spruce, a wood of interest, are tested with different thicknesses of 19 mm and 50 mm with equal other dimensions. The surface is monitored by a well-known system designed by FORTH, DHSPI for around 3 days.

The interferograms are reconstructed in the pre-processing step and the obtained images are named the wrapped phase. This step is followed by the processing including the denoising of the wrapped phase images and the unwrapping of these images. The denoising includes a median filter followed by a Sine–Cosine Average with Stationary Wavelet Transform (SCA-SWT) algorithm. The unwrapped phase is realised with the Calibrated Phase Unwrapping based on Least-Square and Iterations (CPULSI) algorithm with its MatLab function. The next step, called post-processing is the calculation and display of the deformation map. All these steps can be executed for all the acquired data of one surface mock-up in one row.

From the acquired data, a raw interferogram, also called the wrapped phase, can be reconstructed. In the studied case, a set of images is taken every five minutes and a first pattern could be identified for these five minutes after reconstruction. This raw wrapped phase is denoised by a 2D median filter with symmetric padding and a square window size that have to be at least $3 \times 3$ to denoise enough and not more than $4 \times 4$ to keep all the details. The time of computing is slightly increasing with the increase of the size of the window. A second algorithm, called SCA-SWT, is applied to the already denoised images to have a better signal-to-noise ratio. This method uses a stationary wavelet transform on the sine and cosine denoised interferograms with a five-level decomposition and a db2 or sym2 analysing wavelets that are chosen for their sharp form and fast processing time. This transform is followed by the application of a Birgé-Massart threshold with a default sigma and an alpha equal to 6.25. The inverse transform of the sine and the cosine are calculated before the denoised wrapped phase reconstruction. Some drawbacks due to the image extension can appear on the border of the image. The wrapping process can be performed by two different algorithms depending on what one is looking for. The PUSLI method can give us rapidly the absolute variation or displacement of one image useful for giving an alert. The CPULSI with a longer calculation time will give the real value and orientation of the displacement. Both of them are based on the least-squares method that needs a suitable limit of iteration chosen at 300 and an error to reach that is given at 0.001. The CPULSI method uses calibration with a phase-known point, this point is chosen as the point in the denoised wrapped phase with a value nearer to zero. Finally, the deformation map can be visualised as a 2D or 3D map.

The results show that there is a displacement of 2 μm that can be visualised. It may not be significant but if it accumulates through cycles, it can have an important impact on artworks. It can cause a loss in the plasticity of the different materials [26] involving defects apparition. In addition, this relative change on the scale of the whole surface can be the first signal of the inhomogeneous response to the environmental changes that may damage the artwork.

## 6. Conclusions

In this study, a novel approach is proposed to process the DHSPI raw data to make the procedure quicker and more possible to be automatic, since the quantity of data from this technique rises rapidly. The proceeding of data treatment is executed on the raw data as the pre-processing, denoising, and unwrapping to obtain deformation maps of the recorded surface.

With this approach, the change of the surface of the wooden panel can be calculated, and this deformation is the response to the relative humidity variations, which can help to monitor the artworks in the museum. The DHSPI used with a passive approach and an adapted cycle of tracking can allow a regular following, depending on the artwork's typology, on the state of the artwork's materials. The DHSPI could also be used for checking before and after displacements for temporary exhibitions, loans, restoration processes, and authentification.

Future works will be focused on using deep learning [27] to replace the processing step. This may fasten the proposed processing and surely improve it. From this data processing can be extracted points on the surface that moves more than the others. Monitoring these

points on artworks could help to assess preventive conservation in museums. Indeed, not usual movements of these points could be detected and a threshold could be created to define artworks degradations and needs more careful attention.

**Author Contributions:** Methodology, X.B.; Validation, M.A.; Writing—original draft, J.A.–L.S.; Supervision, V.D., N.W.-C. and V.T. All authors have read and agreed to the published version of the manuscript.

**Funding:** The experimental work was conducted at the Holography-Metrology laboratory at the Institute of Electronic Structure and Laser/Foundation for Research and Technology-Hellas (IESL/FORTH) of the Ultraviolet Laser Facility supported in part by the European Union's Horizon 2020 research and innovation programme LASERLAB-EUROPE (Grant Agreement No. 654148 or 871124). This research was funded by Fondation des Sciences du Patrimoine grant number anr-17-eure-0021.

**Institutional Review Board Statement:** Not applicable.

**Informed Consent Statement:** Not applicable.

**Data Availability Statement:** Not applicable.

**Acknowledgments:** This work was supported by the école universitaire de recherche psgs hch humanities, creation, heritage.

**Conflicts of Interest:** The authors declare no conflict of interest.

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
