# Peer review of "Surface Displacement Measurements of Artworks: New Data Processing for Speckle Pattern Interferometry"

_applsci, doi:10.3390/app122311969_

Round 1

Reviewer 1 Report

Manuscript Title: Surface displacement measurements of artworks: new data processing for 1 speckle pattern interferometry

Manuscript ID: applsci-1974140

Journal: Applied Sciences

The paper will have a greater contribution toward protecting precious heritage from the impact of environmental fluctuations.

Abstract- The abstract shall be structured in one paragraph. Separate paragraphs aren’t required. It is a precise summary of the research work indicating the purpose/ objective of the research, methods and materials employed, major findings/ conclusion drawn from the study and implications. However, in this abstract, the major findings of the study are overlooked.

Debug some grammatical or mechanical errors in the abstract section. For instance, “The control systems chosen by museums depend on the size and the age of the 18 building and on financial means”. Remove the preposition on highlighted in red. And, look at line 19…there is a lack of methods (consider parallelism). Either there is a lack of method or there are lacks of methods.

The manuscript had better include a section entitled “Literature Review” to conceptualize and develop the paper properly. It also helps the readers to grasp the issues or cases from various heritage sites.

 2. Theory of evaluated data processing …This heading can be modified as Model Specification of Data Processing.

Results and Discussions

-          The authors should substantiate their findings with recently published scholarly works (2018, 2019, 2020, 2021, and 2022). They need to show whether the findings of the current research are consistent with previously published related works.

-          The practical implications of the paper are not detailed. The authors shall clearly indicate how this paper contributes to policymakers, conservationists and curators regarding heritage conservation and management.  

Author Response

We would like to thank you for your questions and your remarks that enabled us to improve the content of our article. Please found our answers in the attachment.

Reviewer 2 Report

This investigation is of great importance, taking into account the control of energy expenditure, as well as the preservation of the built heritage. Concerning the paper and in my opinion:

- The figures 1 and 2 should be removed from the introduction and insert in an additional section regarding material and methods to be used in the paper.

- The introduction should introduce the different parts of the paper, and aims should be clarified

- The references should be uniformized between them (e.g., references 3,5,11,12, and 13).

- The section 2.1. pre-processing should be better framework, considering the different and subsequent methods presented, and the added value for the research “cultural heritage” in this case museum rooms, reinforced with references.

- Also, section 2.2. Denoising methodologies should be sustained with references.

- Figure 3 - should better be introduced in the text and explained.

- Improve the resolution, if possible, in figures 5 and 6.

- In my opinion the legends from figures 6, 7, 8, 9, 10, 11 and 12 should be explained in the main text of the paper and the legends should be shortened.

- In my opinion should be introduced a section with the discussion of the different methods and phases of the process and its relevance for the aims of the paper.

-The conclusion section should meet the abstract and introduction aims (e.g., how curators should develop preventive conservation strategies and have tried to control temperature and relative humidity in the museum rooms to stabilise artworks.

With these changes I believe the paper can be published.

Author Response

(The authors gave the same response as above.)

Reviewer 3 Report

89: Is data processing optimization aimed at increasing processing speed, accuracy, or some other parameter?

The presented scheme of the experiment does not allow us to evaluate the obtained results.

What was the range of relative humidity inside the chamber during the experiment? What device and how it was controlled?

Was the initial humidity of the test samples the same?

How the samples were illuminated by the laser (continuous or pulsed, power/energy densities on the surface of the samples) and how the registration was carried out (I mean the laser and the camera were outside the chamber, the stability of the laser radiation parameters over time, whether the heating of samples and air by radiation was taken into account, fluctuations of air from for heating, vibration effect, etc.)

Сonclusions do not contain results for the impact of the volume of wooden support on the possible degradation

The presented interferograms were obtained from the entire surface of the samples (100 * 100 mm) or from some part?

Do the obtained values of changes of about 2 microns significantly affect the preservation of art objects?

The conclusions  duplicate the content of the article and they need to be generalized

Author Response

(The authors gave the same response as above.)

Round 2

Reviewer 1 Report

The authors tried to modify the manuscript as per the comments. Hence, it can be published once the galley proof is over.  

Reviewer 3 Report

The article can be accepted in present form